# Vaccine Production Process: How Much Does the General Population Know about This Topic? A Web-Based Survey

**DOI:** 10.3390/vaccines9060564

**Published:** 2021-05-29

**Authors:** Angela Bechini, Paolo Bonanni, Beatrice Zanella, Giulia Di Pisa, Andrea Moscadelli, Sonia Paoli, Leonardo Ancillotti, Benedetta Bonito, Sara Boccalini

**Affiliations:** 1Department of Health Sciences, University of Florence, 50134 Florence, Italy; angela.bechini@unifi.it (A.B.); paolo.bonanni@unifi.it (P.B.); beatrice.zanella@unifi.it (B.Z.); leonardo.ancillotti@stud.unifi.it (L.A.); benedetta.bonito@unifi.it (B.B.); 2Medical Specialization School of Hygiene and Preventive Medicine, University of Florence, 50134 Florence, Italy; giulia.dipisa@unifi.it (G.D.P.); andrea.moscadelli@unifi.it (A.M.); sonia.paoli@unifi.it (S.P.)

**Keywords:** vaccine manufacturing, immunization, hesitancy, pharmaceutical industry, quality control, knowledge, online survey, safety

## Abstract

Background: Vaccine hesitancy has been recognized as a major global health threat by the World Health Organization. Many studies have investigated vaccine safety as a determinant for vaccine hesitancy; however, not much attention has been paid to vaccine production and quality control during the vaccine production process or whether knowledge about this topic may influence vaccine confidence. The aim of this study was to characterize the common knowledge about the vaccine production process. Methods: A freely accessible online questionnaire was developed on Google Modules and disseminated through social networks. A descriptive analysis of the collected answers was performed, and the chi-square test was used to assess significant differences for the sociodemographic characteristics of the study population (age, gender, work or education and training in the healthcare setting, minor offspring). A binary logistic regression model was performed considering these socio-demographic categories as independent variables. Results: The number of collected questionnaire was 135. Most of the participants (127/135, 94.1%) were aware that quality control measures are carried out during manufacturing, although some knowledge gaps emerged in specific aspects of the vaccine production process, without statistically significant differences between age groups. Working in the healthcare setting or being educated in healthcare may be considered predictors for a better understanding that more than 50% of the production time is spent on quality control (AOR = 3.43; 95% CI: 1.84–8.14, *p* = 0.01) and that considering quality control performed during the vaccine production process is adequate for avoiding contamination (AOR = 7.90; 95% CI: 0.97–64.34; *p* = 0.05). Conclusions: This study allowed for a characterization of common knowledge about the vaccine production process. It highlighted the need to implement specific strategies to spread correct information about the vaccine production process. This study may contribute to increased confidence and trust in vaccines and vaccination among the general population.

## 1. Introduction

Vaccines and vaccinations are invaluable resources for health protection, both for the individuals and for the community. Vaccines are cost-effective preventive tools that have allowed significant goals to be achieved in public health during the last few decades [1,2,3]. Recently, the World Health Organization (WHO) estimated that vaccinations prevent up to three million deaths each year [4]. High vaccination levels allow the incidence of vaccine-preventable diseases (VPD) to be reduced and, if possible, herd immunity to be attained [5]. 

Despite these accountable benefits, vaccine hesitancy has gradually increased among the general population, representing one of the top ten global health threats in 2019 [6]. Vaccine hesitancy, defined as ‘*the delay or refusal in the administration of vaccines, despite the availability of safe and effective vaccines and vaccination services*’, is a complex issue that varies across time and geographical contexts; it may also be driven by the different types of available vaccines and by many other factors, including complacency, convenience, and confidence [7]. For a consistent segment of the population, the benefit/risk balance of vaccination has shifted toward it being a risk, in particular due to fear of an adverse event following immunization (AEFI). The possibility of experiencing an AEFI has reduced vaccine acceptance [5,8]. Thus, it is currently challenging to spread the benefits of vaccines and vaccinations in the apparent absence of their related infectious diseases, although an adverse event may occur (even if generally mild and transitory) [9].

Vaccine safety is one of the main determinants that contribute to vaccine hesitancy [7], despite vaccines having to meet the highest safety and quality standards. Indeed, safety is one of the primary issues for any vaccine, which is assessed throughout every step of vaccine development (during preclinical and clinical trials) as well as after its licensure [10]. Furthermore, health authorities require ongoing and strict commitment toward post-licensure safety assessment through passive and the active vaccine surveillance systems. The scientific literature has extensively analyzed the relationship between vaccine hesitancy and vaccine safety [11,12,13,14,15,16,17,18,19], while not much attention has been focused on vaccines production and quality control during the vaccine production process. In particular, the knowledge (or beliefs) of the general population about the vaccine production process could influence their trust and confidence toward available vaccines. Indeed, some studies suggested that greater knowledge of vaccine-preventable diseases and their related vaccines (such as influenza or HPV) may increase vaccine confidence and, thus, vaccine acceptance [20,21].

The vaccine production process, as for all medical drugs, must comply with Good Manufacturing Practices (GMPs); moreover, it must undergo more quality control steps during production than any other drug. The production of a vaccine usually takes 6–24 months; however, the entire process, including all development phases, may sometimes take several years to be completed due to the requirement of strict quality control and quality assurance mechanisms [22,23]. Quality tests have been developed and validated in order to verify conformity to stringent specifications for purity (such as the purity of the cell substrate used for each production cycle), quality of the microorganism harvest, adequacy of all phases in the production process, and the safety and potency of the final bulk vaccine filled in the final containers [24]. The pharmaceutical company must comply with this strict recommendation, and the manufacturer has the primary legal responsibility for the safety, quality, and efficacy of the products marketed [25].

Although the high quality and safety levels required throughout the vaccine production chain could have a significant positive impact on people’s trust and vaccination adherence, this issue has not yet been extensively investigated. Therefore, our aim was to characterize knowledge about the vaccine production process among the general population using an online survey.

## 2. Materials and Methods

### 2.1. Questionnaire Development and Web Dissemination

A cross-sectional study was carried out to investigate the knowledge and opinions of the general population about the vaccine production process through the dissemination of an anonymous online ad hoc questionnaire, developed using the Google Modules application by a group of experts in the field. The survey was previously tested in a pilot study on a small sample of volunteers (15–20 subjects) [26,27]; inappropriate questions were accordingly changed. The survey was spread through the social network Facebook and the website Vaccinarsintoscana (https://www.vaccinarsintoscana.org/ accessed on 9 February 2021), and it was available from January 2020 to May 2020. Moreover, the participants could share the questionnaire with their contacts via social platforms or instant messaging applications within this same period.

The only inclusion criterion was to provide consent to participate in the study. The questionnaire was addressed to subjects aged ≥18 years, internet users, and social media users. The study was conducted according to the guidelines of the Declaration of Helsinki [28].

The survey was made up of two sections:Section 1: Sociodemographic information, including age groups (20–29 years, 30–39 years, 40–49 years, >49 years), sex, work or education and training in the healthcare setting (work/education: HC; work/education: not HC), and offspring (minors/non-minors);Section 2: Information about knowledge on the vaccine production process.

Section 2 of the questionnaire included multiple choices or “YES/NO” questions and one open question (short open answer) about what participants would like to see or ask if they were visiting a pharmaceutical company. The questionnaire is available in the Appendix A.

### 2.2. Descriptive and Statistical Analysis of Collected Answers

The answers were automatically collected into a database and subsequently analyzed using IBM SPSS Statistics 25 (IBM Corp. Released 2017. IBM SPSS Statistics for Windows, Version 25.0. Armonk, NY, USA: IBM Corp). The enrolled population was classified into different groups according to the sociodemographic information: age, sex, type of work or education (healthcare and not healthcare), and offspring (minors and non-minors). A descriptive analysis was carried out to describe the main characteristics of the participants and to assess the frequencies and the percentages of the collected answers for the different sociodemographic groups. A chi-square test was used to assess significant differences in the answers considering the sociodemographic characteristics of the participants [26,27]. A *p*-value less than or equal to 0.05 was considered statistically significant. 

### 2.3. Binary Logistic Regression Model

The likelihood that some variables can predict a specific answer was tested by applying a logistic regression model considering gender, age, professional profiles in healthcare settings, and having offspring as independent variables. We applied the “Enter Method” for entering variables into the multivariate logistic regression model. Questions with yes/no answers were implemented in a binary logistic regression model, and the questions with multiple answers were all encoded as “right/wrong” answers in the same regression model [29,30,31]. The odds ratio was adjusted for all the independent variables, resulting in an adjusted odds ratio (AOR).

The Hosmer–Lemeshow goodness-of-fit statistics (H–L test) were used to assess whether the model adequately described the data [32].

## 3. Results

The number of collected questionnaires was 136; one participant did not give consent to the use of the questionnaire data. Thus, we analyzed 135 surveys. 

### 3.1. Sociodemographic Characteristics of the Study Population

A total of 88 females (65.2%) and 47 males (34.8%) participated in the survey. The youngest participant was 21 years old, while the oldest participant was 69 years old. The mean age was 35.4 years, and most of the participants were 20–39 years old (73.4%). In this study population, 47 participants worked or had an education in the healthcare setting (34.8%); among these, 32 were females (68%) and 15 were males (31%). About 72% of the participants did not have minor children (Table 1).

### 3.2. Knowledge about Vaccine Production Process

Figure 1 summarizes the overall answers collected in Section 2 of the questionnaire; the distribution of knowledge and opinions about vaccine production process stratified by socio-demographic characteristics is shown in Appendix A. 

Almost all of the participants (127/135, 94.1%) thought that controls were carried out during the production process. With respect to the type of work or education, all participants who worked or had a healthcare education were aware of control activities carried out during the vaccine production process (47/47) compared to 90.9% (80/88) of subjects who did not work or have an education in the healthcare field. About 44.4% of the participants (60/135) answered that the vaccine production process is more controlled than that of common drugs, while 52.6% thought that the controls were the same (71/135). Only a few participants (4/135, 3%) thought that the vaccine production process was less controlled than that of other drugs. 

Concerning the amount of time needed to produce a vaccine, 66% (89/135) of the subjects answered 6–24 months (giving the right answer), while 27.4% (37/135) answered 3–5 years, and only 6.7% (9/135) answered less than 2 months; no statistical difference was found for the different groups. Regarding the time spent in quality controls during the vaccine production process, 62.2% (84/135) of participants answered more than 50% of the time (giving the correct answer), 34.1% (46/135) believed 10–50% of the time, and 3.7% (5/135) thought less than 10% of the time. About 79% (37/47) of subjects who worked or had education in the healthcare setting answered more than 50% (giving the correct answer), compared to 53.4% (47/88) of subjects who did not work or have an education in the healthcare field (*p* = 0.02). 

About 58% (78/135) of participants thought that quality controls were fulfilled by all members involved in both production and distribution (National Health Institute, the drug company, and external laboratories). Significant differences were observed in the answers given by subjects who had a job/education in the healthcare field and those who did not. Within the first group, about 68% (32/47) of subjects thought that quality controls were carried out by the National Health Institute, drug company, and external laboratories (giving the right answer), whereas, in the second group, the percentage decreased to 52% (46/88), representing a statistically significant difference (*p* = 0.03). Almost half of the participants (67/135, 49.6%) thought that quality controls were carried out during each phase of vaccine production, even during the transport and storage phases (giving the correct answer). No significant difference was observed between groups.

A slightly higher percentage of the sample (75/135, 55.6%) supposed that a vaccine cannot be contaminated by impurities during the production, whereas most participants (121/135, 89.6%) affirmed that adequate controls were carried out to avoid contamination. A significant difference in the frequencies of the answers was observed between subjects who worked or had an education in the healthcare field and subjects who did not (affirmative answer: 97.9% and 85.2%, respectively; negative answer: 2.1% and 14.8%, respectively; *p* = 0.02) (Appendix A).

Lastly, most of the participants (107/135, 79.3%) affirmed that the marketed/licensed vaccines had the same characteristics as those studied before production and clinical trials. Significant differences were found between females and males (73.9% and 89.4% of women and men, respectively, answered “yes”; 26.1% and 10.6%, respectively, answered “no”; *p* = 0.03) and between subjects who had minor children and those who did not (68.4% of those who had minor offspring and 83.5% of those who did not answered “yes”; 31.6% and 16.5% answered “no”, respectively; *p* = 0.05) (Appendix A).

Furthermore, we analyzed participants’ answers with the respect to age group. The analysis highlighted a certain degree of heterogeneity in the percentages of the given answers.

In some cases, higher percentages for the correct answers were found in subjects aged 20–29 and 30–39 compared to older age groups (Appendix A). In particular, younger participants more frequently gave the correct answers to the following questions: “During the vaccine production process, how much time is spent on quality controls?”, “Who controls the vaccine production process?”, and “When is the quality of vaccine controlled?” Nevertheless, no significant differences were observed among the age groups for any question.

### 3.3. Participants’ Requests If They Were To Visit a Pharmaceutical Company

The last question of the survey was an open question to collect information about what participants would like to see or ask if they could visit a pharmaceutical company. We collected 62 relevant answers (removing five items not related to the theme), some of which had multiple explanations (Table 2). Most of the answers were about the vaccine production process in general terms (e.g., “I would like to see how a vaccine is produced.” or “I would like to see each phases of the production.”) or about specific phases of the process (e. g., “I would like to see how to inactivate viruses and bacteria, such as the purification phase or the storage of materials used in the production process.”). The second most represented theme was related to quality control and the types of tests performed, followed by the possibility of visiting laboratories and the research development sector. Lastly, 11 participants expressed indifference toward visiting a pharmaceutical company.

### 3.4. Predictors of Previous Knowledge or Opinions about Vaccine Production Process 

A multivariate logistic regression model was developed to assess the predictors for the answers of the questionnaire. Table 3 shows only the questions for which almost one of the independent variables (sex, age, work or education and training in the healthcare setting, and minor offspring) resulted significant (*p* ≤ 0.05).

We found that working or having an education in the healthcare setting was a predictor factor for knowing that more than 50% of the production time is spent on quality control (AOR = 3.43; 95% CI: 1.84–8.14; *p* = 0.01) and for considering the control performed during the vaccine production process as adequate in order to avoid contamination (AOR = 7.90; 95% CI: 0.97–64.34; *p* = 0.05). In addition, sex could be considered a predictor factor. Indeed, we found that males seemed to be more inclined to believe that licensed vaccines had the same characteristics of those previously studied (AOR = 4.26; 95% CI: 1.32–13.75; *p* = 0.05).

The Hosmer and Lemeshow test confirmed the satisfactory GOF (goodness-of-fit) for our model (0.662; 0.873; 0.763) (Table 3).

## 4. Discussion

### 4.1. Assessment of Participants’ Knowledge toward Vaccine Production 

The results from our survey highlighted that almost all the participants were aware that controls are carried out during the vaccine production process (94.1%), that quality controls are adequate to avoid contamination with impurities (about 90%), and that licensed vaccines have the same characteristics as those tested during vaccine development process (about 80%). This good level of awareness among our respondents regarding the quality controls performed during the vaccine production process seems to be in line with the results of a review published in 2020, which highlighted that most European citizens know that vaccines are strictly tested [33]. On the other hand, we found some knowledge gaps about specific aspects of the vaccine production process. For example, about 53% of participants thought that vaccines underwent equal quality control to other drugs, whereas it should be clear that the vaccine production process requires a higher number of quality and assurance controls than any other drug [22,34].

A consistent segment of our participants (about 38%) did not know that a significant amount of the whole vaccine production process (50–70%) is spent completing all the quality control testing analyses [34]. It is estimated that major manufacturers generally run 100–500 different quality control tests throughout the course of vaccine production, and they constantly assess the safety, potency, and purity of their products [24,25]. 

Quality control testing is based on the general requirements for biologicals drafted by the WHO and National Control Authorities (NRAs); thus, the NRAs play a key role in assuring product quality as they are responsible for the review of licensing applications, lot release, and monitoring the performance of the product. In Italy, the Italian Agency of Medicines (*Agenzia Italiana del Farmaco*—*AIFA*) and the Italian National Institute of Health (*Istituto Superiore di Sanità*—ISS) follow all quality control phases during vaccine production by carrying out recurring inspections at the laboratories and at the manufacturing plants. Along with their activities, external certified and accredited laboratories also perform quality assessments and release a certificate of conformity [35,36,37]. Most of participants (about 58%) were aware that the Italian National Institute of Health, the drug company, and external laboratories oversee the quality assessment of the final vaccine products. Nevertheless, about 42% of interviewees thought that controls were carried out only by one of these figures. Moreover, the timing for carrying out quality controls was partially known by our participants; about 48% of the interviewed subjects thought that quality controls were performed at different stages of the production or, alternatively, during transportation and storage. As a matter of fact, quality controls are carried out throughout all steps, according to the standards established by international and national authorities: during the vaccine production cycle and before the distribution of each batch on the market [37]. 

More than half of the participants (55.6%) thought that vaccines cannot be contaminated by impurities during production. The adequacy of vaccine production and the possible presence of contaminants due to the production process are checked twice for each batch before its delivery to the vaccination centers [37]. Different analyses are performed to assess purity (i.e., chromatography, mass spectrometry, and electrophoresis) and the presence of bacteria, such as mycoplasma, endotoxins, or other microbial contaminants (microbiological assay and polymerase chain reaction) in the vaccine formulation throughout the whole production process [24,25]. Another important aspect considered to minimize the risk of contamination is the vial composition; for example, the anti-meningococcal vaccine ACWY does not contain preservatives among its components. Thus, it is not suitable for multidose vials as this may increase the risk of contamination after opening [25,38]. Although vaccine production process must comply with GMPs, some incidents many occur, as happened for a Chinese biotechnology company that specialized in vaccine production. Indeed, it was found to have violated good manufacturing practices and then supplied less effective vaccines in China. This fact significantly increased doubts on vaccination among caregivers who knew about this incident [39].

Our questionnaire also investigated participants’ point of view about the correspondence between licensed vaccines and those tested during the clinical trials in terms of biological characteristics. Interestingly, about one-fifth of the interviewees (20.9%) believed that licensed/marketed vaccines might differ from those studied in the trials. As a matter of fact, the last goal of analytical quality control testing during the vaccine production process is to guarantee that the product leaving the manufacture is equivalent to that described on the registered label [25].

### 4.2. Sociodemographic Characteristics as Predictors of Acquired Knowledge

Considering the collected answers related to the sociodemographic characteristics of our sample, statically significant differences were found in the percentages of correct answers between subjects who worked or had an education in the healthcare setting and those who did not. As expected, we generally retrieved higher percentages for correct answers among subjects who worked or had an education in the healthcare setting. Indeed, the logistic regression model highlighted that working or having an education in the healthcare setting may be considered a predictive factor. Subjects who worked or had an education in the healthcare setting were positively associated with correctly knowing how much time is spent on quality control during the vaccine production process (OR = 3.43; 95% CI: 1.84–8.14) and with considering that quality controls ensure that vaccines are not contaminated by impurities (OR = 7.90; 95% CI: 0.97–64.34). Thus, we can assume that participants who worked or had an education in the healthcare setting could have previously acquired some information about this issue. Our findings may reflect the efforts carried out in recent years to increase healthcare workers’ knowledge about vaccines and vaccination in terms of safety and quality issues. Some studies highlighted how vaccine hesitancy is also a concern with respect to healthcare workers [40,41,42,43,44]; thus, it was recognized as a priority to implement tailored interventions in order to increase vaccination knowledge and confidence among primary healthcare workers [43,45]. For example, structured health interventions, such as educational lectures, considerably increase the knowledge level among healthcare personnel [46]. Our results seem to be in line with the European scenario, whereby healthcare professionals are generally confident about the safety and the importance of vaccination and have higher levels of confidence than the general population [47].

Nevertheless, some gaps in knowledge about specific aspects of the vaccines production process were also found among those who had an education or worked in the healthcare setting; this may represent an attitudinal or behavioral barrier towards healthcare workers’ immunization uptake [48].

Moreover, we found that males seemed to be more inclined to believe that licensed vaccines had the same characteristics as those previously studied compared to females. A similar result was discussed in the last report of the “Vaccine Confidence Project”, in which women were found to be less likely than men to have high overall confidence in vaccines and vaccination in some European countries (Austria, Croatia, Czech Republic, Estonia, France, Greece, Italy, Lithuania, Luxembourg, Romania, and Slovakia) [47].

Age and having minor offspring were not found to be predictive factors for acquiring knowledge about the vaccine production process. No data are available in the literature about this issue; nevertheless, considering the European context, older people (over 65 years old) have higher confidence in vaccination than the younger population. Moreover, in some European countries (Ireland and Slovenia), subjects with children have a higher level of vaccine confidence, whereas in Denmark, Romania, and Sweden, higher confidence was described for people without offspring. No data are available for Italy or other European countries [47].

### 4.3. Participants’ Interests in Increasing Their Knowledge

Lastly, the analysis of the open answers related to things participants would like to ask or see if they were to visit a drug manufacturer highlighted the need for our study population to acquire more information about the vaccine production process and the tests applied for quality control assessment. This result may also reflect a need for the general population. Thus, health authorities and stakeholders should consider the possibility of setting up meetings or similar events to increase the general population’s knowledge about the vaccine production process. The implementation of this educational session may also be helpful to reduce vaccine hesitancy. In the current situation of the COVID-19 pandemic, the interest of the population toward vaccine development and production has increased and it is essential to correctly inform the population about this topic to avoid the rise and dissemination of fake news on vaccine safety and quality. This seems particularly relevant for the Italian scenario, as a recent systematic review suggested a low or suboptimal level of COVID-19 vaccine propensity to get vaccinated among the Italian general population (ranging from 53.7% to 77.3%) compared to other countries [49].

### 4.4. Strengths and Limitations of the Study

The main strength of our study is the novelty of the topic investigated, which involved characterizing people’s knowledge on vaccine production in relation to some socio-demographic characteristics. However, our study had some limitations. First, our study population included a limited number of participants, representing a small sample of convenience. As a consequence, having collected zero-frequency for some responses may have affected the *p*-value. This limited participation may have been due to the period in which the survey was launched, i.e., at the beginning of the pandemic. Therefore, the public’s attention was mainly focused on this new health emergency and less on other health topics, such as vaccines and vaccination. Moreover, subjects aged 20–39 years were mostly represented compared to older subjects. We should consider that the collected answers may not be entirely representative of the general population. Although the use of a web-based survey allowed us to easily reach a greater number of subjects, this may have resulted in a possible sampling bias due to the exclusion of the non-digitalized population and less frequent users of social media. 

## 5. Conclusions

The scientific literature has widely discussed the concerns about vaccine hesitancy and vaccine safety; however, not much attention has been paid to vaccine production process and whether this may influence people’s attitudes toward vaccines and vaccination. The aim of this paper was to characterize the people’s knowledge on the vaccine production process. To our knowledge, this is the first study to investigate the level of awareness and the opinions acquired by the population regarding the vaccine production process. 

People seem to be aware that quality controls are carried out during the vaccine production process; nevertheless, some aspects of the production process are poorly known or completely unknown. Moreover, those who worked or had an education in the healthcare setting had a greater knowledge about this issue. This finding supports the importance of introducing training interventions about vaccines and vaccinations targeted at healthcare personnel.

Considering the period in which the survey was launched, it would be interesting to carry out a new study in a post-pandemic period to see if the knowledge and understanding of vaccine development has increased. A recently published study has highlighted that trust in the healthcare system or in vaccine manufacturers are both positively associated with acceptance of the COVID-19 vaccine [50].

In the future, in order to address the lack of knowledge among the general population concerning the vaccine production process, health authorities should consider the need to implement specific strategies for spreading information about this topic. This may result in an increase in awareness about vaccines and, thus, in better vaccination compliance and confidence.

## Figures and Tables

**Figure 1 vaccines-09-00564-f001:**
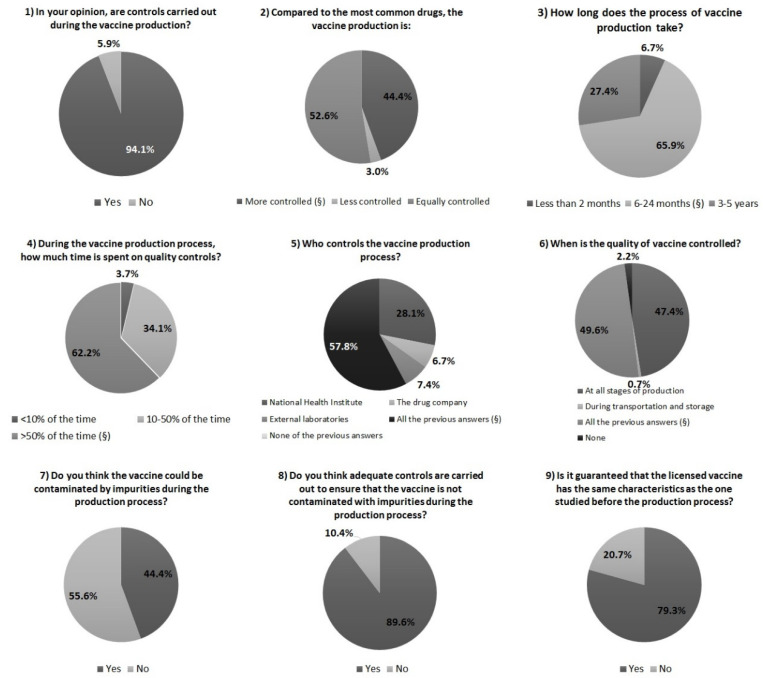
Results of Section 2 of questionnaire: summary of the distribution of knowledge and opinions about vaccine production. Note: §—correct answer.

**Table 1 vaccines-09-00564-t001:** Summary of the sociodemographic data of the study population (N = 135).

Sociodemographic Characteristics of the Enrolled Participants			
		n	% (n/N)
**Age groups (years)**	20–29	48	35.6
	30–39	51	37.8
	40–49	18	13.3
	>49	18	13.3
**Sex**	Male	47	34.8
	Female	88	65.2
**Work or education in the healthcare setting**	Yes	47	34.8
	No	88	65.2
**Minor offspring**	Yes	38	28.1
	No	97	71.9

**Table 2 vaccines-09-00564-t002:** Summary of the collected answers for the open question: “If one day you could visit a pharmaceutical industry, what would you like to ask or to see?” (N = 62).

“If One Day You Could Visit a Pharmaceutical Industry, What Would You Like to Ask or to See?”	
	n (%)
**The whole vaccine production process or specific phases**	29 (46.8)
**Quality control**	15 (24.2)
**Laboratories and Research & Development**	5 (8.1)
**Regulatory aspects**	3 (4.8)
**Research studies, efficacy studies**	3 (4.8)
**Production timing**	2 (3.2)
**Company structure (managers and workers)**	2 (3.2)
**Costs related to vaccine’s production**	2 (3.2)
**Time spent in training**	1 (1.6)

**Table 3 vaccines-09-00564-t003:** Multivariate logistic regression analysis. Note: β—regression coefficient, AOR—Adjusted Odds Ratio.

During the Vaccine Production, How Much Time is Spent on Quality Controls?					
PREDICTOR FACTORS	β	OR	SE	95%CI	*p*-Value
AGE					
20–29 years	−0.341	0.71	0.60	0.22–2.31	0.57
30–39 years	0.208	1.23	0.63	0.36–4.23	0.74
40–49 years	−0.235	0.79	0.73	0.19–3.29	0.75
>49 years	Reference group	-	-	-	-
SEX					
Male	0.002	1.00	0.40	0.45–2.21	1.00
Female	Reference group	-	-	-	-
TYPE OF WORK/EDUCATION					
Healthcare	1.232	3.43	0.44	1.44–8.14	**0.01**
Not Healthcare	Reference group		-	-	-
OFFSPRING					
Minors	0.220	1.25	0.48	0.48–3.22	0.65
Non minors	Reference group	-	-	-	-
	significance value H–L test = 0.662				
**Do you think adequate controls are carried out to ensure that the vaccine is not contaminated with impurities during the production process?**					
	β	OR	SE	95%CI	*p*-value
AGE					
20–29 years	0.268	1.31	0.99	0.19–9.13	0.79
30–39 years	−0.347	0.71	0.97	0.10–4.76	0.72
40–49 years	−0.593	0.55	1.06	0.07–4.42	0.58
>49 years	Reference group		-	-	-
SEX					
Male	0.456	1.58	0.68	0.42–5.97	0.50
Female	Reference group	-	-	-	-
TYPE OF WORK/EDUCATION					
Healthcare	2.067	7.90	1.07	0.97–64.34	**0.05**
Not Healthcare	Reference group	-	-	-	-
OFFSPRING					
Minors	0.082	1.085	0.70	0.28–4.26	0.91
Non minors	Reference group	-	-	-	-
significance value H–L test = 0.873					
**Is it guaranteed that the licensed vaccine has the same characteristics as the one studied before the production?**					
	β	OR	SE	95%CI	*p*-value
AGE					
20–29 years	0.741	2.10	0.76	0.47–9.37	0.33
30–39 years	0.745	2.11	0.78	0.46–9.74	0.34
40–49 years	−0.112	0.89	0.84	0.17–4.61	0.89
>49 years	Reference group	-	-	-	-
SEX					
Male	1.450	4.26	0.60	1.32–13.75	**0.02**
Female	Reference group	-	-	-	-
TYPE OF WORK/EDUCATION					
Healthcare	0.500	1.65	0.52	0.60–4.56	0.34
Not Healthcare	Reference group	-	-	-	-
OFFSPRING					
Minors	−0.759	0.47	0.54	0.16–1.35	0.16
Non minors	Reference group	-	-	-	-
	significance value H–L test = 0.763				

## Data Availability

Data sharing not applicable. Data were collected and managed in aggregated form according to European Union Regulation 2016/679 of the European Parliament and the Italian Legislative Decree 2018/101.

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
