# Peer review of "Vaccine Production Process: How Much Does the General Population Know about This Topic? A Web-Based Survey"

_vaccines, 2021, doi:10.3390/vaccines9060564_

Round 1

Reviewer 1 Report

The paper entitled " Vaccines’ production process: how much does the general population know about this topic? A web-based survey " is an interesting study that aimed to improve the understanding about the knowledge on the vaccines’ production process among the general population through an online survey.

The manuscript is well written and well supported by consistent references. The conclusions are largely sound and improve the existing knowledge. Therefore, I think that the paper has enough quality to be published in Vaccines.

Some suggestions to improve the manuscript are reported below.

  • Row 124. Replace gender with sex
  • Table 1. The authors report the total number of participants for each variable, instead they should report it at the end of the table caption e.g (N = 135).
  • Table 1 has two captions, authors should delete one. “Table 1. Summary of the sociodemographic data of the study population. “ “Sociodemographic characteristics of the enrolled participants”
  • Paragraph 3.2 "Knowledge about vaccines' production process" the authors should delete the following sentence because it is a repetition: "Section 2 of the questionnaire included nine questions with" YES / NO "or multiple-choice answers".
  • The table 2 has an unclear / explanatory caption “Results of Section 2 of questionnaire: overall answers and related to the different groups”. The caption should specify that the table summarizes the distribution of knowledge about vaccines' production process stratified by socio-demographic characteristics.
  • Authors should take into account the high frequency of zero-frequency responses because this can affect the p-value, so they risk considering significant differences that may not be.
  • Table 4: Authors should rewrite the caption like this: ”Summary of the collected answers for the open question: If one day you could visit a pharmaceutical industry, what would you like to ask or to see? Furthermore, together with the absolute frequency they should calculate the relative frequency. The N should be replaced with the lowercase n.
  • Row 221 replace "Table 5 showed" with "Table 5 shows"Gli autori dovrebbero specificare se il OR è aggiustato, e per quali variabili.
  • The discussion paragraph should begin with a brief summary of the main findings.
  • The discussion from lines 237 to 243 is more appropriate for the Introduction paragraph.
  • Row 248. It is not clear what results the authors are discussing.
  • From row 259 to 280 the authors pay too much attention to the quality control management, I do not think it is appropriate to discuss it in this part of the manuscript.
  • The Authors often repeat the results in the Discussion rather than comparing them to other studies or trying to explain them.
  • The authors should discuss the results highlighting that a part of the participants (about 1/3) who work or education in the healthcare setting did not answer correctly.
  • Finally, the authors should eliminate repetition of results in the Conclusions paragraph

Reviewer 2 Report

Abstract

  • “however, not much attention has been given to vaccines’ production and quality control during the vaccines’ production process.” Why? Is your aim to evaluate the impact of subjects’ knowledge about the production of vaccine and quality control on vaccine hesitancy? Please clarify.
  • Methods: Who developed? Validation procedure? When was developed? Inclusion and exclusion criteria of participants? How many questions? What data were collected? Statistical methodology?
  • Results: How many participants (total)? How many participants with education in the healthcare setting? “between age groups” – What age groups?
  • Conclusion: the conclusion should reply to study aim, which is “to improve the understanding about the people’s knowledge on the vaccines’ production process.” Please rewrite conclusion (or study aim).

Keywords: please browse some MeSH terms: https://www.ncbi.nlm.nih.gov/mesh/; Ideally, keywords should be different from the words of title and abstract.

Introduction

  • The first paragraph is too long.
  • Please cite some systematic reviews about vaccine hesitancy. Please see: https://pubmed.ncbi.nlm.nih.gov/?term=vaccine+hesitancy&filter=pubt.systematicreview and https://www.ncbi.nlm.nih.gov/pmc/?term=(%22Vaccines%22%5BJournal%5D)+AND+vaccine+hesitancy
  • These reviews should also be cited/explained in Discussion.
  • The topic of vaccine hesitancy during COVID-2019 pandemic should be addressed. Please see “COVID-19 Vaccine Hesitancy Worldwide: A Concise Systematic Review of Vaccine Acceptance Rates” https://www.ncbi.nlm.nih.gov/pmc/articles/PMC7920465/ or Confidence and Receptivity for COVID-19 Vaccines: A Rapid Systematic Review https://www.ncbi.nlm.nih.gov/pmc/articles/PMC7823859/
  • Supposition: people’s knowledge on the vaccines’ production process will reduce vaccine hesitancy. Please present some evidence. The introduction should support the interest/relevance of the study.
  • “Therefore, our purpose was to improve the understanding about the knowledge on the vaccines’ production process among the general population through an online survey.” or “to characterize the knowledge on the vaccines’ production process among the general population through an online survey.” I am not sure about study objective.

  1. Materials and Methods

- Materials and methods are too compact; please present more subheadings.

- Materials and methods should be exhaustive and reproductible. Please give more details/information.

- Who developed the questionnaire? What experts? How were experts selected? How the experts developed the questionnaire? Validation procedure? Protocol? Decision process? How have the experts agreed about the questions? Why 15-20 subjects in the pilot? References to support 15-20 subjects? Positive and negative findings of pilot? Type of study? Settings? When/Where/How was the questionnaire developed? Exclusion criteria of participants? Statistical methodology? References of studies with similar statistical methodologies? Ethical approval? How have you checked the age of participants? All adults? How were the veracity of the sociodemographic data checked (e.g., age or health education)? Calculation of sample size (n=135)? Have you applied a formula? Similar studies? Confidentiality of data?

- Please present the questionnaire in methods (not in a supplementary file).

- How was the open question scored? Criteria/protocol? Who scored this question? Please see guidelines about qualitative studies here https://www.equator-network.org/reporting-guidelines-study-design/qualitative-research/

- Were findings double checked? Quality control?

- Were requisites of the application Chi-Square tests verified?

- Who validated the binary logistic regression model? Has a statistician been involved?

- How was the binary logistic regression model validated?

- Were data from 135 questionnaires sufficient to develop a logistic regression model?

- For instance see: Validation of Logistic Regression Models in Small Samples: Application to Calvarial Lesions Diagnosis https://www.sciencedirect.com/science/article/pii/S0895435698001656 and Validation techniques for logistic regression models https://pubmed.ncbi.nlm.nih.gov/1925153/.

- If necessary, please consult a statistician.

  1. Results

3.2 Knowledge about vaccines’ production process

- Please use some illustrations/graphics to present study findings.

- Please restructure the presentation of data from Table 2 and 3. It seems you have presented the bulk results of crosstabulations, which is not very usual. Please try to present data in another format (more comprehensible). Results must be more intelligibly presented.

3.3 Participants’ requests if they were in visit to a pharmaceutical industry.

- Table 4 – please present %s

- Were the guidelines of qualitative studies followed? Please give more details about the application of these guidelines. Please see https://www.equator-network.org/reporting-guidelines-study-design/qualitative-research/

3.4 Predictors of previous knowledge or opinions about vaccines’ production process

- Please present the outputs of the validation of the model. It seems the sample size is too short to develop a logistic regression model. Or not?

- “How do you validate a regression model? Methods to determine the validity of regression models include comparison of model predictions and coefficients with theory, collection of new data to check model predictions.”

- Please see Validation of Regression Models: Methods and Examples https://www.tandfonline.com/doi/abs/10.1080/00401706.1977.10489581

  1. Discussion

- Discussion is too compact; please use more subheadings (e.g., at least one subheading per each section of results)

- Please cite more references in discussion. For instance, cite references to explain each one of the explanatory variables of the regression model.

- Please check if all study findings are discussed.

- Please discuss study strengths and weakness.

- Please present a section about study limitations, practical implications, and future research directions at the end of discussion.

- Please discuss possible study biases in study limitations. Please see: Identifying and Avoiding Bias in Research. https://www.ncbi.nlm.nih.gov/pmc/articles/PMC2917255/ Is sample size representative?

  1. Conclusion

- Study conclusions should reply to study objectives. Please rewrite study conclusions.

Reviewer 3 Report

Vaccines-1184837

Title: Vaccines’ production process: how much does the general population know about this topic? A web-based survey.

Response: Thank you for the opportunity to review this project. I found it interesting.

Major issue: I found the categories of ‘worked and in healthcare confusing’. In the table it was described as ‘work/education: HC and Work/education, not HC’. But here in this description it says: Taking into account…..’all the participants who worked or had a healthcare education were……  of the subjects who did not work or had an education in the healthcare field. The differences in these two groups is different narratively from that in the tables, and same confusion in the results and discussion.

I am having a hard time with the comparison, then the whole study if one compares those who work and had a healthcare education to those who don’t work or had a healthcare education. I do think that the comparison of those who work, those who don’t, and those who had a healthcare education is valid enough to study.

Abstract: Page 1; lines22-25. Rewrite for clarity in English

Page 1-2: The whole first paragraph is too long, including many thoughts throughout. Suggest dividing this paragraph into three paragraphs. First paragraph vaccinations as a preventive measure. Second paragraph to include vaccine hesitancy. Third paragraphs to include vaccine safety vs hesitancy. Also, there are many minor grammatical issues within this whole section.

 Last paragraph of the Introduction is professionally written and clear.

Page 2, line 92. Change: to provide the consent to the study …..  to provide the consent ‘in’ the study.

Page 3, line 129. Add to ‘minors’ with children or minor offspring.

Page 4, lines 138-141. I found the categories of ‘worked and in healthcare confusing’. In the table it was described as ‘work/education: HC and Work/education, not HC’. But here in this description it says: Taking into account…..’all the participants who worked or had a healthcare education were……  of the subjects who did not work or had an education in the healthcare field. The differences in these two groups is different narratively from that in the tables, and same confusion in the results and discussion.

Page 5: lines 181-185. Please include the discussion that there were far more participants in the 20-29 and 30-39 age group than in the older groups. The point that the younger groups had more correct answers is then not a point at all.

Page 11; lines 215-216. I would suggest that this be rewritten as: Lastly, 11 participants chose they do not know or not interested in visiting pharmaceutical industry.

Page 11; lines 221-222. Table 5 sentence needs clarity. When the authors write ‘for which almost one significant predictor was retrieved’ it sounds like only one factor may have been significant.

Page 12; table 5. Please explain ‘Reference groups’ in this table.

Page 14; line 323. Please change ‘ad’ to ‘had’.

Page 15; line 375. Change who worked or had ‘and’ education…..  to Change who worked or had ‘an’ education

Overall, this is an interesting study. There are quite a few limitations to the validity of the study considering the lack of older participants, the lack of clarity of the HC, worked or not worked definitions, convenience sampling, and timing of the survey. It would be interesting to conduct another survey post pandemic to see if the knowledge and understanding of vaccine development has increased. The grammatical errors are throughout the manuscript, I would suggest having an editor help with the clarity would be beneficial.

The analysis was appropriate and impressive. The question arises from the clarification of the categories studied.

The areas of concern need to be addressed.

Thank you for allowing me to review this manuscript.

Round 2

Reviewer 2 Report

In general, authors have accepted all suggestions. Many Thanks. As required a Proofreading certificate has been presented.

Some commentaries:

  • Please update methods, point 2.3 Descriptive and Statistical Analysis of Colleted Answers:
  • Please see Ranganathan, P., Pramesh, C. S., & Aggarwal, R. (2017). Common pitfalls in statistical analysis: Logistic regression. Perspectives in clinical research, 8(3), 148–151. https://doi.org/10.4103/picr.PICR_87_17

  • Results, point 2.1, line 105: The study was conducted according to the guidelines of the Declaration of Helsinki. A reference is missing.
  • Point 2.2, line 119: please check the citation of SPSS. “IBM Corp. Released 2017. IBM SPSS Statistics for Windows, Version 25.0. Armonk, NY: IBM Corp.”?
  • Point 2.3: “2.3. Binary Logistic Regression Model”. References are missing. Please check if the results are correctly presented, please see:

  • Kalil, A. C., Mattei, J., Florescu, D. F., Sun, J., & Kalil, R. S. (2010). Recommendations for the assessment and reporting of multivariable logistic regression in transplantation literature. American journal of transplantation : official journal of the American Society of Transplantation and the American Society of Transplant Surgeons, 10(7), 1686–1694. https://doi.org/10.1111/j.1600-6143.2010.03141.x

  • Zhang, Y. Y., Zhou, X. B., Wang, Q. Z., & Zhu, X. Y. (2017). Quality of reporting of multivariable logistic regression models in Chinese clinical medical journals. Medicine, 96(21), e6972. https://doi.org/10.1097/MD.0000000000006972

  • Point 3.2: lines 157 to 213; please present some graphics. The paper will benefit from the presentation of some figures in results. Please see diverse examples of Vaccines journals with figures here: https://www.mdpi.com/search?q=vaccine+hesitancy&journal=vaccines
  • Point 3.4: Lines 260-261 “The Hosmer and Lemeshow test confirmed the satisfactory GOF (goodness-of-fit) for our 260 model.” Please present the values and/or refer that these values are presented in Table 5.

  1. Discussion

- Discussion is too compact, please create more subheadings.

- You may consult papers from Vaccines journal with subheadings in discussion here:

  • https://www.mdpi.com/search?q=vaccine+hesitancy&journal=vaccines

Please consider these two examples of papers from Vaccines journal about vaccine hesitancy with figures and subheadings in discussion:

  • Attitudes towards COVID-19 Vaccination among Hospital Staff—Understanding What Matters to Hesitant People

  • What Is the State-of-the-Art in Clinical Trials on Vaccine Hesitancy 2015–2020?

  • I have more one recommendation: please update discussion with studies about patient knowledge/opinion related to the manufacturing of vaccines. For instance, see:

  • Wong, M., Wong, E., Huang, J., Cheung, A., Law, K., Chong, M., Ng, R., Lai, C., Boon, S. S., Lau, J., Chen, Z., & Chan, P. (2021). Acceptance of the COVID-19 vaccine based on the health belief model: A population-based survey in Hong Kong. Vaccine, 39(7), 1148–1156.

“Multivariable regression analyses revealed that perceived severity, perceived benefits of the vaccine, cues to action, self-reported health outcomes, and trust in healthcare system or vaccine manufacturers were positive correlates of acceptance”

  • Geoghegan, S., O'Callaghan, K. P., & Offit, P. A. (2020). Vaccine Safety: Myths and Misinformation. Frontiers in microbiology, 11, 372. https://doi.org/10.3389/fmicb.2020.00372

“Although parents and patients have a number of concerns about vaccine safety, among the most common are fears that adjuvants like aluminum, preservatives like mercury, inactivating agents like formaldehyde, manufacturing residuals like human or animal DNA fragments, and simply the sheer number of vaccines might be overwhelming, weakening or perturbing the immune system.”

  • Du, F., Chantler, T., Francis, M. R., Sun, F. Y., Zhang, X., Han, K., Rodewald, L., Yu, H., Tu, S., Larson, H., & Hou, Z. (2020). The determinants of vaccine hesitancy in China: A cross-sectional study following the Changchun Changsheng vaccine incident. Vaccine, 38(47), 7464–7471.

  • Impact of incidents with the production of vaccines on patients.

Reviewer 3 Report

Authors,

This manuscript is much improved overall, with the inclusion of the survey (very helpful). The written English improvements helped in understanding the study. 

The only item I am concerned about is the survey responses as yes/no. This does not allow the the real variances of the participants responses. Having used The Lichert style survey would have captured these variances. I would suggest if there is a follow up 'post-pandemic' survey, the Lichert scale be used. 

I appreciate reviewing this manuscript again. 
